# Rare Clinical Symptoms in Hairy Cell Leukemia: An Overview

**DOI:** 10.3390/cancers16173054

**Published:** 2024-09-01

**Authors:** Tadeusz Robak, Marcin Braun, Agnieszka Janus, Anna Guminska, Ewa Robak

**Affiliations:** 1Department of Hematology, Medical University of Lodz, 91-738 Lodz, Poland; 2Department of General Hematology, Copernicus Memorial Hospital, 93-510 Lodz, Poland; agnieszka_janus@poczta.onet.pl; 3Department of Pathology, Medical University of Lodz, 92-213 Lodz, Poland; marcin.braun@umed.lodz.pl; 4Department of Nuclear Medicine, Copernicus Memorial Hospital, 93-513 Lodz, Poland; anna.m.guminska@gmail.com; 5Department of Dermatology, Medical University of Lodz, 90-647 Lodz, Poland; ewarobak@onet.eu

**Keywords:** hairy cell leukemia, leukemia cutis, pulmonary involvement, bone lesions, neurological manifestations, pulmonary symptoms, ocular manifestations, cardiac manifestation, rare symptoms

## Abstract

**Simple Summary:**

Diagnosis of hairy cell leukemia is based on the presence of hairy cells in bone marrow and peripheral blood as well as the characteristic immunophenotype. Moreover, in classic HCL, most patients present with a *BRAF* V600E mutation. The typical symptoms of classic hairy cell leukemia include pancytopenia, massive splenomegaly and increased risk of infection. However, rarer manifestations of HCL are occasionally reported, including cutaneous symptoms, bone infiltration, arthritis and central nervous system symptoms, as well as gastrointestinal tracts, heart, lungs, ocular involvement and other symptoms.

**Abstract:**

Background: Hairy cell leukemia (HCL) is a rare indolent B-cell lymphoid malignancy. The majority of patients are asymptomatic and HCL is usually diagnosed incidentally during a routine blood cell count. In symptomatic patients, typical symptoms are related to pancytopenia and splenomegaly. In this review, we present rare clinical symptoms in patients with HCL. Methods: A literature search was conducted of PubMed, Web of Science and Google Scholar for articles concerning hairy cell leukemia, leukemia cutis, bone lesions, neurological manifestations, pulmonary symptoms, ocular manifestations, cardiac manifestation and rare symptoms. Publications from January 1980 to August 2024 were scrutinized. Additional relevant publications were obtained by reviewing the references from the chosen articles. Results: Extramedullary and extranodal manifestations of classic HCL are rare. However, leukemic involvement in the skin, bone, central nervous system, gastrointestinal tract, heart, kidney, liver, lung, ocular system and other organs have been reported.

## 1. Introduction

Classic hairy cell leukemia (HCL) is a rare, indolent B-cell leukemia that makes up about 2% of all cases, with an annual incidence of about 0.3 cases per 100,000 [1,2,3,4,5]. HCL occurs more predominantly in middle age, with a median age of diagnosis of 56 years (range: 40–70 years) [1,2]. The disease is extremely rare in younger patients; however, a few cases of HCL have been reported in children [6]. Diagnosis is based on peripheral blood (PB) and bone marrow (BM) morphology, flow cytometry, immunophenotyping, immunohistochemistry and molecular studies [1,2,7,8]. The majority of patients with HCL are asymptomatic at diagnosis and the disease is usually found incidentally during routine blood cell counts. The typical clinical presentation of classic HCL involves pancytopenia, monocytopenia, splenomegaly, constitutional symptoms and increased risk of infection [1,2]. In most patients, lymph nodes are not enlarged; however, an atypical presentation, such as the absence of splenomegaly and the presence of lymphadenopathy and leukocytosis, may also be observed in some patients. Typical hairy cells are medium in size with moderately abundant pale blue cytoplasm with a characteristic serrated cytoplasmic border, reniform nuclei, open chromatin and absent nucleoli [1,2,9,10,11,12,13]. The characteristic immunophenotype of classic HCL cells includes the co-expression of CD19, CD20, CD11c, CD25, CD103, CD200, FMC7 and CD123. The unique criterion for the diagnosis of HCL is the co-expression of CD103, CD25 and CD11c. Also, HCL can be distinguished from other B-cell lymphomas, including HCLv, based on annexin A1 expression; indeed, it has been reported that Annexin A1 is a 100% specific immunohistochemical marker for classic HCL [7]. The *BRAF* V600E mutation is present in almost all HCL patients and plays a key role in the pathogenesis of this disease [8]. Extramedullary and extranodal manifestations of classic HCL are rare events [9,10]. However, leukemic involvement in the skin, bone, central nervous system, gastrointestinal tract, heart, kidney, liver, lung, ocular system and other organs have been reported [10,11,12,13] (Table 1).

HCL variant (HCLv) is a clinical–pathologic entity with intermediate features between classic HCL and B-cell prolymphocytic leukemia [92,93,94,95,96,97,98]. In the fifth edition of the WHO classification (WHO-HAEM5), classic HCL is classified as splenic lymphoma/leukemia, together with splenic marginal zone lymphoma (SMZL) with circulating villous cells in peripheral blood (PB), splenic lymphoma with a prominent nucleolus (SLPN) and splenic diffuse red pulp lymphoma (SDRPL) [94]. In this classification, HCLv is included as SLPN together with CD5-negative B-prolymphocytic leukemia (B-PLL). However, HCLv still exists in the clinical International Consensual Classification (ICC) [95,96]. Cladribine and pentostatin have been the drugs of choice for more than 30 years for newly diagnosed HCL [4,99,100,101]. In relapsed patients, re-treatment with purine analogs, alone or in combination with rituximab, is still an acceptable option. However, re-treatment with single-agent purine analogs often results in shorter responses, with some patients becoming refractory to the drugs. For these patients, other treatment options can be considered, including interferon-α (IFN-α), rituximab, bendamustine, fludarabine and BRAF inhibitors, used alone or in combination [4,98].

This review presents some of the rare clinical symptoms reported in patients with classic HCL. A literature search was conducted of PubMed, Web of Science and Google Scholar for articles concerning hairy cell leukemia, leukemia cutis, bone lesions, neurological manifestations, pulmonary symptoms, ocular manifestations, cardiac manifestation and rare symptoms. Publications from January 1980 to August 2024 were scrutinized. Additional relevant publications were obtained by reviewing the references from the chosen articles.

## 2. Skin Symptoms

Cutaneous manifestations of HCL have been reported in about 10–12% of patients [14]. Skin symptoms may be specific to HCL or mostly non-specific, due to autoimmune reactions, infections and secondary neoplastic or drug-induced reasons. Specific cutaneous manifestations, known as leukemia cutis, are very rare and are defined as the direct infiltration of the epidermis, the dermis and the subcutaneous tissue by leukemic cells [14,15,16,17]. They can be manifested as disseminated erythematous maculopapules or nodules, ranging from violaceous to red-brown in color, or flesh-colored nodules, with central ulceration or infiltrative eruptions [14,15,16,17]. Skin changes are localized in one region or are disseminated in several parts of the skin, and are characterized microscopically as a perivascular mononuclear leukemic cell infiltration. In patients with HCL, the diagnosis of leukemia cutis is performed based on the histopathology of skin biopsy and the immunophenotyping of neoplastic cells [17,18]. Skin biopsy and immunophenotyping must be performed in all patients with suspicion of leukemia cutis. Leukemia cutis should be differentiated from other cancers, vasculitis, infections and inflammation [19,20].

In most presented cases, specific treatment of HCL resulted in the disappearance of cutaneous infiltrates. Vasculitis is one of the more common non-specific skin changes in HCL patients [21,22,23], and vasculitis syndromes such as cutaneous leukocytoclastic vasculitis (CLCV), polyarteritis nodosa (PAN), pyoderma gangrenosum and paraneoplastic vasculitis have been noted in HCL and other lymphoid malignancies. Vasculitis can present as the initial manifestation of HCL or can be diagnosed in the course of disease [21,22,23,24]. Moreover, vasculitis may be a reaction to infection or a paraneoplastic syndrome related to HCL itself [25]. In one report of 42 HCL patients, CLCV was identified in 21 and PAN in 17 [26]. Neutrophilic dermatoses like Sweet’s syndrome, pyoderma gangrenosum and neutrophilic eccrine hidradenitis are rarely observed in HCL.

Sweet’s syndrome (acute febrile neutrophilic dermatosis), characterized by erythematous painful lesions that are sometimes in the form of plaques, has been reported as a presenting symptom at diagnosis or at relapse of HCL [27,28]. Pyoderma gangrenosum is a neutrophilic, reactive, non-infectious and inflammatory dermatosis involving the skin and mucosal tissue. It has been reported in a few cases with HCL. Pyoderma gangrenosum was successfully treated with cladribine without other immunosuppressive drugs such as cyclosporine or corticosteroids [102,103,104]. Elsewhere, Elkon et al. describe four patients with HCL and systemic vasculitis similar to polyarteritis nodosa that developed within two years of the onset of HCL [105], while Zervas et al. present a case of leukemia-associated polyarthritis [106].

Due to the increased risk of infections, the most common skin involvement in patients with HCL is due to bacterial or viral infections. However, verrucae, dermatophyte, candidal infections and pyogenic infections (i.e., abscesses, cellulitis, folliculitis and pyoderma) were also observed in some patients. Opportunistic skin infections, including atypical mycobacterial skin infections, fungal infections and Ecthyma gangrenosum have also been reported in HCL patients [14]. In most patients, prognosis is not affected by skin symptoms [14,15].

## 3. Bone Lesions

The most common orthopedic complications in HCL are due to osteolytic and osteoblastic lesions, severe osteoporosis, aseptic necrosis of the femoral head and multifocal lytic changes [29,30,31,32,33,34,35,36,37,38,39,40,41,42,43,44]. Pathological femoral fractures and asymptomatic or painful diffuse osteosclerosis with extensive thickening of bone were also reported in some patients [35,36,37].

Skeletal infiltrations are observed in approximately 3% of HCL patients [38,39,40,41], and some patients present with skeletal infiltration at diagnosis [32,42]. However, in most patients, bone involvement is recognized between a few weeks and several years after diagnosis of HCL [38]. The median time from diagnosis of HCL to the manifestation of skeletal symptoms is 20 months (range: 0–93), although some patients have developed skeletal symptoms in complete hematological remission, and skeletal involvement has been found to develop 22 years after HCL diagnosis [31]. Some HCL patients with lytic bony lesions did not demonstrate splenomegaly [43,44].

Diagnosis and treatment evaluation of bone lesions is based on computed tomography (CT) and positron emission tomography (PET) (Figure 1.) [38]. Before the age of 18F-FDG PET/CT imaging, skeletal lesions were most commonly diagnosed by CT and magnetic resonance imaging (MRI), which were also used to evaluate the response to treatment. MRI is indicated in patients who have no roentgenographic or bone scan alterations, or they are non-specific [39,107]. More recently, 18F-FDG PET/CT imaging was recommended for diagnosis, staging and treatment assessment (Figure 1) [31,108]. PET is more specific and sensitive than other imaging methods, including CT and MRI, and is more helpful in differentiating HCL lesions from malignant bone tumors. Prognosis is usually not affected by bone leukemic infiltration [30,31]. To sum up, bone lesions in HCL can be early or appear over the years, and even in patients in remission.

## 4. Pulmonary Symptoms

Pulmonary involvement by HCL cells is a very rare condition (Figure 2), with only a few cases with leukemic lung infiltration being apparent in the literature [45,46,47,48,49,50,51,109]. A lung biopsy for histological examination is indicated in patients with pulmonary symptoms if antibiotics and antifungal treatment are not effective. However, a pulmonary biopsy is connected with the risk of bleeding and traumatic lung. Treatment with cladribine and/or rituximab induced a regression of leukemic pulmonary involvement in most patients [45,46,47,48,49,50]. However, the most common cause of pulmonary symptoms in HCL are infections [45]. Disseminated atypical mycobacterial infections were reported in several patients with HCL [51,109,110,111]. Clinically, pulmonary infiltrations were usually observed on chest X-rays in addition to fever and chills. Invasive diagnostic studies, including thoracotomy, can be required for the confirmation of the diagnosis of atypical mycobacterial infection with *Mycobacterium kansasii*, *Mycobacterium avium-intracellulare* and *Mycobacterium chelone*. Mediastinal lymphadenopathy and secondary pulmonary amyloidosis was also observed in some patients with HCL [10,51].

The concomitant occurrence of lung cancer and HCL were occasionally reported. In some patients, a *BRAF*-V600E-mutated lung adenocarcinoma coexisted with *BRAF*-mutated HCL [112,113]. These patients can be treated with BRAF-targeted therapy, including vemurafenib and dabrafenib.

## 5. Neurological Manifestations

Neurological complications were reported in approximately 5% of HCL patients [52], with the most common cause being infection. However, the direct invasion of leukemic cells to the central nervous system (CNS) is rare in HCL [52,53,54,55,56,57,58,59,60,61,62,63]. In HCL patients, CNS involvement is typically indicated by confusion, aphasia, headache, meningeal syndrome, motor ataxia, dizziness, weakness, confusion, slurred speech, frequent falls, facial droop, fatigue, blurry vision and acute delusional symptoms [53,54,55,56,57,58,59,60,61,62]. Studies have also reported HCL localization in brain parenchyma and/or meninges [53,57,58,62,63,64,65,66,67,68]. In most HCL patients, CNS involvement is unrelated to large cell lymphoma disease transformation unlike CLL [59,63,64,65]. Systemic therapy with purine analogs and CD20 antibodies or, more recently, with BRAF inhibitors, are usually effective in controlling HCL demonstrating CNS involvement [53,60,63,66].

## 6. Ocular Symptoms

Ocular involvement of HCL is extremely rare and only 12 cases have been reported so far [69,70,71,72,73,74]. Of the reported cases, most involved the globe, with only two affecting the orbital mass [74,75]. In three-fourths of the patients, ocular or orbital manifestations were identified at the diagnosis of HCL.

Panuveitis and retinal vasculitis coincident with HCL related to leukemic infiltration or bleeding have been noted in case reports [71,73,76], as has leukemic severe left-sided panuveitis with conjunctival and ciliary involvement in an HCL patient [76]. Local and systemic antibiotic therapy was ineffective, but rapid improvement was observed following treatment with cladribine.

Spontaneous subperiosteal orbital hematoma was also reported in an HCL patient with a history of orbital floor reconstruction due to orbital fracture [77]. The patient received cladribine monotherapy. Four months later, the clinical symptoms (proptosis and diplopia) had resolved and a near-complete resolution of the subperiosteal collection was seen in neuroimaging.

Retinopathy and visual disturbance due to intraretinal hemorrhage were reported as the initial symptom of HCL in a few patients [78,79]. In two cases, leukemic infiltration of the cornea was presented [80]. Also, orbital infections can develop, as HCL itself and antileukemic treatments are highly immunosuppressive. Therapy with cladribine induced an improvement of ocular complications in most patients [71,81].

## 7. Hearing Loss

Sudden hearing loss in patients with leukemia can be related to leukemic infiltration, hemorrhage and infection. El Enazi et al. reported a patient with acute sensorineural hearing loss as an initial manifestation of HCL [114]. The morphology, immunohistochemistry and flow cytometry of the BM confirmed the diagnosis of HCL. This is the only case of sudden hearing loss in HCL noted to date, although a few cases with chronic myeloid leukemia and CLL have been reported [115,116,117,118,119]. Most had unilateral hearing loss, as in the HCL patient.

## 8. Liver and Gastrointestinal Tract Symptoms

Hepatomegaly can be seen in up to 30% of HCL patients [7,120]. However, liver and gastrointestinal tract involvement has only rarely been reported in HCL patients. Dhanesar et al. describe a patient diagnosed with HCL involving the hepatic portal system during post-splenectomy evaluation approximately four years after initial HCL diagnosis [121]. A computed tomography scan of the abdomen showed a tumor in the hepatic portal and peri-portal regions with bile duct obstruction. Following biopsy of the liver mass, a diagnosis of HCL was established [122,123].

Sen et al. describe the first case report of an HCL patient manifesting clinically with duodenal involvement [124]. At HCL diagnosis, the patient had typical symptoms with pancytopenia and massive splenomegaly. A CT scan of the abdomen revealed thickening of the duodenum. Subsequent esophago-gastro-duodenoscopy (EGD) showed duodenal ulcerative inflammation. The duodenal biopsy demonstrated infiltration with HCL cells. Treatment with cladribine leaded to complete response, including the disappearance of duodenum involvement. In another report, Tariq et al. present the case of a patient with gastric cancer that had developed during treatment of hairy cell leukemia with moxetumomab pasudotox [125]. In upper gastrointestinal endoscopy, a gastric mass with irregular margins was observed and histopathological evaluation showed well-differentiated adenocarcinoma of the intestinal type.

## 9. Cardiac Manifestation

Progressive pericarditis and pleuritis in HCL patients at diagnosis was reported recently [126]. The symptoms completely resolved after treatment with purine analogs, with no recurrence observed during a five-year observation. In another report, Koczwara et al. present the case of a 42-year-old HCL patient who developed transient cardiac failure following treatment with cladribine [127]. Cladribine interferes with the cardiac adenylate cyclase pathway and can induce cardiac dysfunction in some patients [128].

At diagnosis, hairy cell leukemia can occassionally mimic infective endocarditis [129]. Ramasamy et al. reported a case of HCL with fever of unknown origin, splinter haemorrhages with vasculitis and moderate splenomegaly and cytopenia at diagnosis, suggesting infective endocarditis; however, the PB cultures and transthoracic echocardiogram were normal, and BM infiltration by CD19, CD25, CD11c, CD45 and CD103-positive lymphoid cells indicated a diagnosis of HCL [129]. Severe congestive heart failure was diagnosed in a patient with HCL treated with recombinant IFN-α2b [130]. Echocardiography showed severe global hypokinesis with second-degree mitral and aortic valve insufficiency and a reduced ejection fraction of 24%. The patient gradually improved after interferon was discontinued.

## 10. Rheumatological Manifestations

Rheumatological symptoms related either to hematological malignancy or disease-related immune dysregulation have rarely been reported in HCL patients [82,83,131]. The first patient with HCL and rheumatoid arthritis (RA) was reported in 1979 by Crofts et al.; in this case, rheumatoid arthritis had developed two years before HCL diagnosis [84]. Westbrook and Golde analyzed 37 HCL patients and identified 6 patients with joint symptoms, usually associated with nodular skin lesions; all patients responded promptly to antileukemic therapy [85]. Facchini et al. reported another patient who developed HCL eight years after RA diagnosis and suggested an association between autoimmune disease and HCL [86]. In another patient, immune RA developed five years after diagnosis of HCL [105]. More recently, Anil et al. reported bilateral knee swelling, erythema, tenderness and mild bilateral pedal edema before HCL diagnosis; the author reported spontaneous resolution of the rheumatism symptoms before anti-leukemic treatment, which indicated that they were immune-mediated [87]. In another case, a patient with HCL developed seropositive RA following treatment with IFN-α. In this case, the RA resolved over the next five months when interferon was administered for the HCL, with no exacerbation of the RA [88].

Zervas et al. presented a patient with HCL who developed seropositive RA [105]. In this patient, the RA resolved when interferon-a treatment was implemented, suggesting that RA with an autoimmune mechanism was directly related to HCL; however, HCL-associated arthritis with a non-immune mechanism has been documented in several patients [82,83,88,89,90,91]. Sattar et al. report the case of a patient with intermittent asymmetrical oligoarthritis who subsequently developed HCL [82]. Importantly, HCL cells were also detected in the synovial fluid, suggesting an association between arthritis and HCL. Raimbourg et al. observed another patient with osteoarticular symptoms among 27 patients with HCL [90]. This patient presented with polyarthritis involving the wrists, ankles and knees, which responded to non-steroidal anti-inflammatory drugs. Hairy cells were identified in the joint fluid, thus establishing a diagnosis of HCL-related arthritis. The presented cases suggest that arthritis can have an immune basis in some patients, but not in others. Prognosis is usually good as antileukemic and immunosuppressive treatment is effective.

## 11. Soft Tissue Involvement

Pilichowska et al. present the case of a patient with primary HCL of the breast, diagnosed incidentally during an elective reduction mammoplasty; the patient was asymptomatic without treatment one year following the HCL diagnosis [132]. In another patient, a pancreatic mass was observed under MRI nine years after HCL diagnosis and treatment with cladribine. An infiltrative lesion was reported along the pancreatic tail: it presented as a pancreatic mass partially encasing the pancreatic body, tail and splenic vessels, with an extension to the splenic hilum [133]. Fine-needle biopsy confirmed HCL infiltration.

## 12. Conclusions

In most patients, HCL is characterized by pancytopenia, splenomegaly and bone marrow infiltration. However, several other unusual symptoms of HCL are reported in the literature. These include extramedullary and extranodal manifestations of classic HCL in various organs including the skin, bones, central nervous system, gastrointestinal tract, heart, kidney, liver, lung and ocular system.

## Figures and Tables

**Figure 1 cancers-16-03054-f001:**
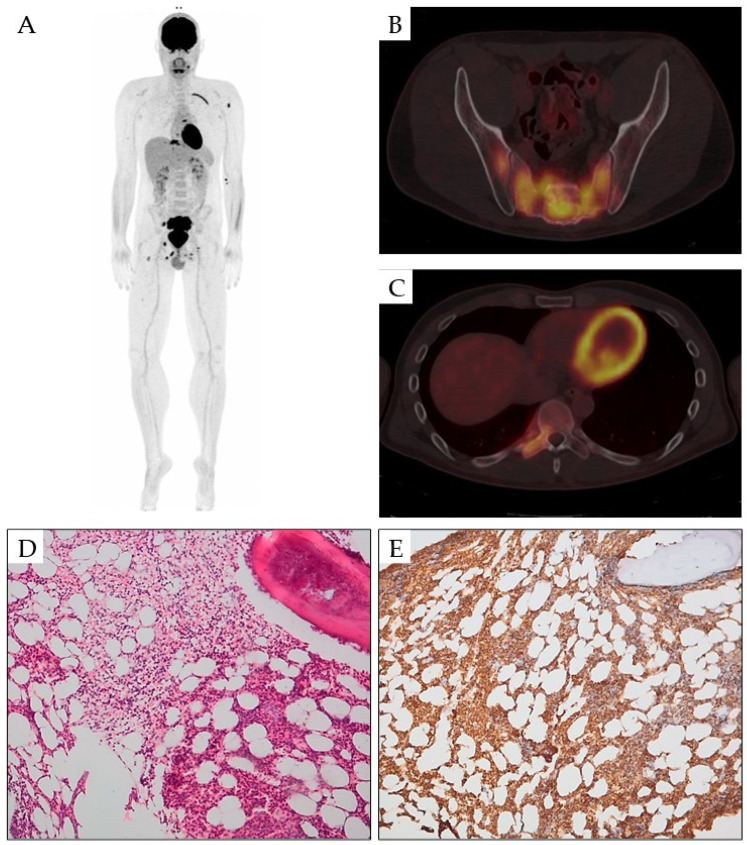
Representative images of hairy cell leukemia bone lesions. Pathologic FDG uptake in the ribs, left humerus, vertebral body Th10, sacrum, right iliac, pubic bone and ischium imaged in PET/CT imaging (**A**). Mixed osteolytic and osteoblastic lesions in the sacrum in PET/CT imaging (**B**) and vertebrum in PET/CT imaging (**C**). Bone marrow infiltrate of HCL in Haematoxyllin and Eosin staining, magnification of 100× (**D**) and in staining for CD20, magnification of 100× (**E**).

**Figure 2 cancers-16-03054-f002:**
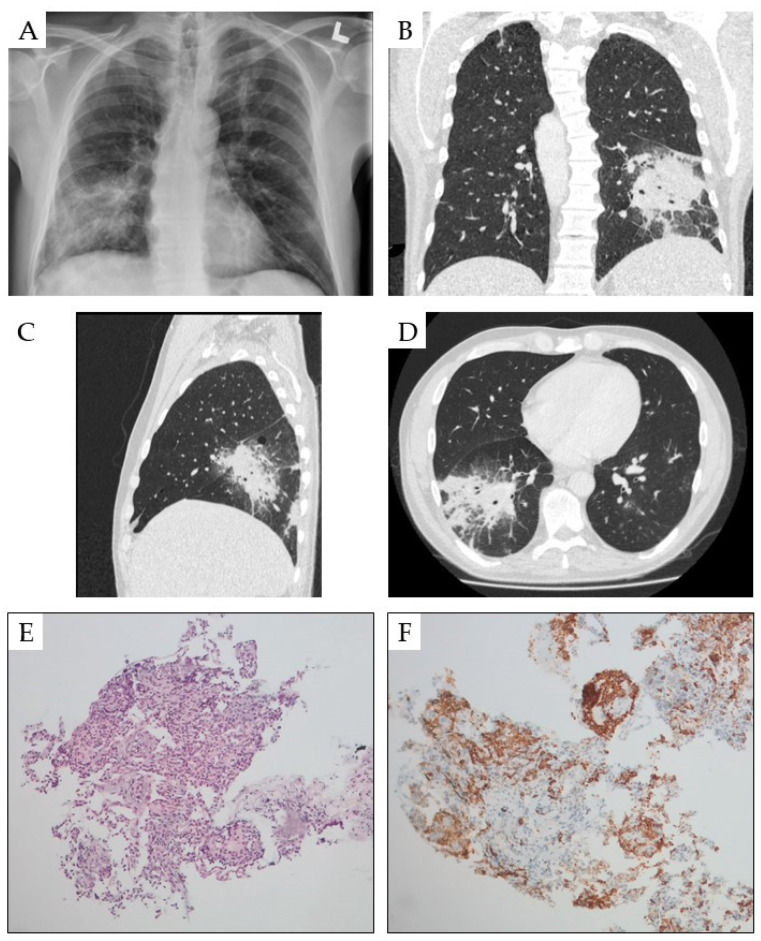
Pulmonary involvement of HCL. X-ray chest PA view shows unilateral right diffuse pulmonary infiltrates (**A**). High-resolution computed tomography reveals septal thickening and multiple small and diffusely-merging focal consolidations with ground glass opacities in frontal (**B**), side (**C**) and transversal projection (**D**). Lung biopsy showing an infiltrate of HCL in Haematoxyllin and Eosin staining, magnification of 100× (**E**) and in staining for CD20, magnification of 100× (**F**). Damage or introduction of infection.

**Table 1 cancers-16-03054-t001:** Rare symptoms in hairy cell leukemia.

Symptoms	Etiology	Clinical Characteristics	Diagnostic Procedures	References
Skin changes	Leukemic infiltration, autoimmune reactions, infections, secondary neoplasms and drug-induced symptoms	Disseminated erythematous maculopapules and nodules, vasculitis, neutrophilic dermatoses and periarteritis nodosa	Skin biopsy	[14,15,16,17,18,19,20,21,22,23,24,25,26,27,28]
Bone lesions	Leukemic skeletal infiltrations	Localized pain, multifocal osteolytic and osteoblastic lesions and severe osteoporosis	X-rays, MRI, CT, PET and core biopsy	[29,30,31,32,33,34,35,36,37,38,39,40,41,42,43,44]
Pulmonary changes	Leukemic infiltration, mediastinal infiltrations and infections	Pulmonary symptoms: cough, dyspnea, chest pain and hemoptysis	Chest X-rays, CT and lung biopsy if antibiotics and antifungal treatment are not effective	[45,46,47,48,49,50,51]
Neurologic manifestations	Leukemic infiltration and infections	Confusion, aphasia, headache, meningeal syndrome, motor ataxia, dizziness, weakness, slurred speech, fatigue and blurry vision	Imaging studies (NMR, CT, PET) and lumbar puncture	[52,53,54,55,56,57,58,59,60,61,62,63,64,65,66,67,68]
Ocular symptoms	Leukemic infiltration, bleeding and infections	Ocular or orbital manifestations, panuveitis, conjunctivitis, leukemic corneal infiltration, retinopathy and visual disturbance	Ophthalmological examination, CT and biopsy of ocular mass	[69,70,71,72,73,74,75,76,77,78,79,80,81]
Rheumatological symptoms	Immune-mediated or direct, non-immune mechanism	Athritis, joint pain and swelling, erythema and tenderness	Serologic tests, X-rays and cytologic evaluation of synovial fluid	[82,83,84,85,86,87,88,89,90,91]

Abbreviations: BM—bone marrow, CNS—central nervous system, CR—complete response, CT—computed tomography, HCL—hairy cell leukemia, MRI—magnetic resonance imaging and PET—positron emission tomography.

## Data Availability

The data presented in this study are available in this article.

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
