# Peer review of "Rare Clinical Symptoms in Hairy Cell Leukemia: An Overview"

_cancers, 2024, doi:10.3390/cancers16173054_

Round 1
Reviewer 1 Report
Comments and Suggestions for Authors
|
The authors reviewed what had been published over approximately 50 years, from 27 January 1980 to August 2024, to capture the incidence of rare clinical conditions associated with HCL. |
|
|
TYPOS |
|
|
1 |
The authors report 'et al' which I suggest always change to 'et al.' |
|
2 |
Line 62: newly-diagnosed Please, change with newly diagnosed |
|
3 |
Line 90 moderately-abundant Please, change with moderately abundant |
|
4 |
Line 103 and line 339: extra nodal Please, change with extra-nodal |
|
5 |
Line 121 Skin Symptoms differentiated with other cancers, Please,change with: differentiated from other cancers, |
|
6 |
line 138 sometimes in form of plaques Please change with: sometimes in the form of plaques |
|
7 |
Line 141 tssue Please change with: tissue |
|
8 |
The roentgenogram (or roentgenograph) is an old-style synonymous term of x-rays |
|
9 |
line 138 sometimes in form of plaques Please change with: sometimes in the form of plaques |
|
10 |
Line 191 and is more useful in the differentiation of HCL lesions and malignant bone tumors. Please change with: and is more helpful in differentiating HCL lesions from malignant bone tumors. |
|
11 |
Line 197 maginification Please change with magnification |
|
12 |
Line 227 is not related Please change with unrelated |
|
13 |
Line 243: due orbital fracture Please change with: due to orbital fracture |
|
14 |
Line 247 Retinopathy and visual disturbance due to intraretinal hemorrhage was reported Please change with: Retinopathy and visual disturbance due to intraretinal hemorrhage were reported |
|
15 |
Line 257 Most of them had Please change with: Most had |
|
16 |
Line 262 has only been rarely reported Please change with: rarely been reported |
|
17 |
Line 281 patient Please, change with: patients |
|
18 |
line 287 can ocassionally mimic infective endokarditis Please change with: can occasionally mimic infective endocarditis |
|
19 |
Lines 294-295 second degree mitral and aortic valve insufficiency and reduced ejection fraction.... Please change with: second-degree mitral and aortic valve insufficiency and a reduced ejection fraction |
|
10 |
Line 305 association Please change with: an association |
|
MINOR COMMENTS
|
|
|
1 |
The methods are mentioned in the abstract but not in the text of the paper where they deserve a section. A PRISMA flow diagram describing the result of the selection process of selected papers can be useful. |
|
2 |
Line 69-Common clinical presentation his paper focuses on the rare clinical features of HCL. This section reports in greater detail the typical clinical characteristics already mentioned in the introduction and in addition, also data about the diagnostic approach and the genetic characteristics of HCL. I would suggest that authors limit this information to the introduction. |
|
3 |
Lines 111-113 Direct infiltration of the skin by leukemic cells 111 is defined as infiltration of the epidermis, the dermis, and the subcutaneous tissue by 112 leukemic cells Lines 117- 119 the diagnosis of leukemia cutis is performed based on the histopatology 118 of skin biopsy and the immunophenotyping of neoplastic cells [27,28]. I would mention the two concepts in the same sentence |
|
4 |
Table 1 is rather chaotic and often unclear. The authors should simplify it by,excluding obvious information, those that cannot be effectively summarized and reporting only the essential and informative points Some typos: Boone Chest X-rey invasion of HCL cells to CNS
|
|
5 |
Line 147 The most common skin symptoms in HCL patients are bacterial or viral infections. I would change with: Due to the increased risk of infections, the most common skin involvement in patients with HCL is due to bacterial or viral infections. |
|
6 |
Lines 155-156: Bone symptoms in HCL include osteolytic and osteoblastic lesions, severe osteopo rosis, aseptic necrosis of the femoral head and multifocal lytic changes I would change to: the most common orthopedic complications in HCL are due to osteolytic and osteoblastic lesions, severe osteoporosis, aseptic necrosis of the femoral head and multifocal lytic changes |
|
7 |
Lines 159-165 The authors can summarize the text by writing that bone lesions can be early, appear over the years and even in patients in remission |
|
|
Pulmonary symptoms. I would limit the report to the rate of patients experiencing the rarest forms of lung infection.I would also mention which treatments are associated with a greater incidence of opportunistic lung infections. Moreover, I would mention the incidence of lung cancers |
|
8 |
Line 253 Sudden hearing loss in patients with leukemia can be related to leukemic infiltration, hemorrhage, and infection. Line 258 Such sudden hearing loss in patients with leukemia can be attributed to leukemic infiltration, hemorrhage, blood hyperviscosity and infec tion The authors should eliminate this repetition |
|
9 |
Line 299 Rheumatological symptoms related either to the hematological malignancy or the immune system have been rarely reported in HCL patients . I would change 'the immune system' with disease-related immune dysregulation Moreover, please change: have been rarely reported. with: have rarely been reported |
|
10 |
I suggest that the authors summarize the text as much as possible, eliminating too detailed descriptions of the reported clinical events. |
Comments on the Quality of English Language
Several typos should be corrected
Author Response
Reviewer 1.
PoczÄ…tek formularza
|
The authors reviewed what had been published over approximately 50 years, from 27 January 1980 to August 2024, to capture the incidence of rare clinical conditions associated with HCL. Response: The authors thank the reviewer for their feedback. |
|
|
TYPOS |
|
|
1 |
The authors report 'et al' which I suggest always change to 'et al.' Response: Corrected as indicated |
|
2 |
Line 62: newly-diagnosed Please, change with newly diagnosed Response: Corrected as indicated |
|
3 |
Line 90 moderately-abundant Please, change with moderately abundant Response: Corrected as indicated |
|
4 |
Line 103 and line 339: extra nodal Please, change with extra-nodal Response: Corrected as indicated |
|
5 |
Line 121 Skin Symptoms differentiated with other cancers, Please,change with: differentiated from other cancers, Response: Corrected as indicated |
|
6 |
line 138 sometimes in form of plaques Please change with: sometimes in the form of plaques Response: Corrected as indicated |
|
7 |
Line 141 tssue Please change with: tissue Response: Corrected as indicated |
|
8 |
The roentgenogram (or roentgenograph) is an old-style synonymous term of x-rays Response: Corrected as indicated |
|
9 |
line 138 sometimes in form of plaques Please change with: sometimes in the form of plaques Response: Corrected as indicated |
|
10 |
Line 191 and is more useful in the differentiation of HCL lesions and malignant bone tumors. Please change with: and is more helpful in differentiating HCL lesions from malignant bone tumors. Response: Corrected as indicated |
|
11 |
Line 197 maginification Please change with maginification Response:Changed as requested |
|
12 |
Line 227 is not related Please change with unrelated Response: Corrected as indicated |
|
13 |
Line 243: due orbital fracture Please change with: due to orbital fracture Response: Corrected as indicated |
|
14 |
Line 247 Retinopathy and visual disturbance due to intraretinal hemorrhage was reported Please change with: Retinopathy and visual disturbance due to intraretinal hemorrhage were reported Response: Corrected as indicated |
|
15 |
Line 257 Most of them had Please change with: Most had Response: Corrected as indicated |
|
16 |
Line 262 has only been rarely reported Please change with: rarely been reported Response: Corrected as indicated |
|
17 |
Line 281 patient Please, change with: patients Response: Corrected as indicated |
|
18 |
line 287 can ocassionally mimic infective endokarditis Please change with: can occasionally mimic infective endocarditis Response: Corrected as indicated |
|
19 |
Lines 294-295 second degree mitral and aortic valve insufficiency and reduced ejection fraction.... Please change with: second-degree mitral and aortic valve insufficiency and a reduced ejection fraction Response: Corrected as indicated |
|
10 |
Line 305 association Please change with: an association Response: |
|
MINOR COMMENTS
|
|
|
1 |
The methods are mentioned in the abstract but not in the text of the paper where they deserve a section. A PRISMA flow diagram describing the result of the selection process of selected papers can be useful. Response: The methods is included in the main text, in the introduction |
|
2 |
Line 69-Common clinical presentation his paper focuses on the rare clinical features of HCL. This section reports in greater detail the typical clinical characteristics already mentioned in the introduction and in addition, also data about the diagnostic approach and the genetic characteristics of HCL. I would suggest that authors limit this information to the introduction. Response: We removed subchapter „Common clinical presentation“ in the revised paper |
|
3 |
Lines 111-113 Direct infiltration of the skin by leukemic cells 111 is defined as infiltration of the epidermis, the dermis, and the subcutaneous tissue by 112 leukemic cells Lines 117- 119 the diagnosis of leukemia cutis is performed based on the histopatology 118 of skin biopsy and the immunophenotyping of neoplastic cells [27,28]. I would mention the two concepts in the same sentence Response: We combined this two sentences in the revised version. |
|
4 |
Table 1 is rather chaotic and often unclear. The authors should simplify it by,excluding obvious information, those that cannot be effectively summarized and reporting only the essential and informative points Some typos: Boone Chest X-rey invasion of HCL cells to CNS Response: Table 1 was reedited according to the rviewer suggestion
|
|
5 |
Line 147 The most common skin symptoms in HCL patients are bacterial or viral infections. I would change with: Due to the increased risk of infections, the most common skin involvement in patients with HCL is due to bacterial or viral infections. Response: Changed as indicated |
|
6 |
Lines 155-156: Bone symptoms in HCL include osteolytic and osteoblastic lesions, severe osteopo rosis, aseptic necrosis of the femoral head and multifocal lytic changes I would change to: the most common orthopedic complications in HCL are due to osteolytic and osteoblastic lesions, severe osteoporosis, aseptic necrosis of the femoral head and multifocal lytic changes Response: Changed as indicated |
|
7 |
Lines 159-165 The authors can summarize the text by writing that bone lesions can be early, appear over the years and even in patients in remission Response: Added as indicated |
|
|
Pulmonary symptoms. I would limit the report to the rate of patients experiencing the rarest forms of lung infection.I would also mention which treatments are associated with a greater incidence of opportunistic lung infections. Moreover, I would mention the incidence of lung cancers Response: Reedited as indicated |
|
8 |
Line 253 Sudden hearing loss in patients with leukemia can be related to leukemic infiltration, hemorrhage, and infection. Line 258 Such sudden hearing loss in patients with leukemia can be attributed to leukemic infiltration, hemorrhage, blood hyperviscosity and infec tion The authors should eliminate this repetition Response: Repetition was eliminated |
|
9 |
Line 299 Rheumatological symptoms related either to the hematological malignancy or the immune system have been rarely reported in HCL patients . I would change 'the immune system' with disease-related immune dysregulation Moreover, please change: have been rarely reported. with: have rarely been reported Response: Changed as indicated |
|
10 |
I suggest that the authors summarize the text as much as possible, eliminating too detailed descriptions of the reported clinical events. Response: The text was reduced as indicated. In particular, Figure 1 and Subchapter „2. Common clinical presentation“ were rmoved. In addition, some sentences were removed from other subchapters |

Reviewer 2 Report
Comments and Suggestions for Authors
The paper by Robak et al. provides a valuable overview of the rare clinical presentations of Hairy Cell Leukemia (HCL), particularly focusing on its extramedullary and extranodal manifestations. This study makes a significant contribution to the field, offering insightful information that could be highly beneficial to hematologists managing HCL cases.
While the paper is comprehensive, it would be advantageous for the authors to elaborate on the clinical relevance of these unusual presentations, particularly regarding their impact on the prognosis of affected patients. Such an addition would enhance the paper's utility for clinicians.
Additionally, Figure 1 seems to primarily depict clinical conditions associated with infectious complications, rather than direct involvement by HCL. Given this discrepancy, the authors might consider removing this figure to maintain the paper's focus.
Author Response
Reviewer 2,
The paper by Robak et al. provides a valuable overview of the rare clinical presentations of Hairy Cell Leukemia (HCL), particularly focusing on its extramedullary and extranodal manifestations. This study makes a significant contribution to the field, offering insightful information that could be highly beneficial to hematologists managing HCL cases. Response: The authors thank the reviewer for their feedback.
While the paper is comprehensive, it would be advantageous for the authors to elaborate on the clinical relevance of these unusual presentations, particularly regarding their impact on the prognosis of affected patients. Such an addition would enhance the paper's utility for clinicians. Response: The prognosis of rarer symtoms was added if available
Additionally, Figure 1 seems to primarily depict clinical conditions associated with infectious complications, rather than direct involvement by HCL. Given this discrepancy, the authors might consider removing this figure to maintain the paper's focus. Response: Figure 1 was removed
